# Metagenomics Insight into Veterinary and Zoonotic Pathogens Identified in Urban Wetlands of Los Lagos, Chile

**DOI:** 10.3390/pathogens13090788

**Published:** 2024-09-12

**Authors:** Catherine Opitz-Ríos, Alvaro Burgos-Pacheco, Francisca Paredes-Cárcamo, Javier Campanini-Salinas, Daniel A. Medina

**Affiliations:** 1Laboratorio Institucional, Universidad San Sebastián, Puerto Montt 5501842, Chile; copitzr1@correo.uss.cl; 2Escuela de Medicina Veterinaria, Facultad de Ciencias de la Naturaleza, Universidad San Sebastián, Puerto Montt 5501842, Chile; aburgosp@correo.uss.cl (A.B.-P.); fparedesc5@correo.uss.cl (F.P.-C.); 3Facultad de Medicina y Ciencia, Universidad San Sebastián, Puerto Montt 5501842, Chile; javier.campanini@uss.cl

**Keywords:** veterinary pathogens, zoonosis, antimicrobial resistance, metagenomics, virulence genes

## Abstract

Wetlands are ecosystems that are essential to ecological balance and biodiversity; nevertheless, human activity is a constant threat to them. Excess nutrients are caused by intensive livestock and agricultural operations, pollution, and population growth, which in turn leads to uncontrolled microbiological development. This impairment in water quality can constitute a risk to animal, human, and environmental health. To thoroughly characterize the microbial communities, shotgun metagenomics was used to characterize the taxonomic and functional pattern of microorganisms that inhabit urban wetlands in the Los Lagos Region of Chile. The main objective was to identify microorganisms of veterinary relevance, assess their potential antibiotic resistance, and characterize the main virulence mechanism. As expected, a high diversity of microorganisms was identified, including bacteria described as animal or human pathogens, such as *Pasteurella multocida*, *Pseudomonas aeruginosa*, *Staphylococcus aureus*, and *Escherichia coli*. Also, a diverse repertory of antimicrobial-resistant genes (ARGs) was detected in metagenomic assembled sequences and inside the sequence of mobile genetic elements, genes that confer mainly resistance to beta-lactams, consistent with the families of antibiotics most used in Chile. In addition, a diverse collection of virulence mechanisms was also identified. Given the significance of the relationship between environmental, animal, and human health—a concept known as One Health—there is a need to establish molecular surveillance programs that monitor the environmental biohazard elements using molecular tools. This work is the first report of the presence of these harmful biological elements in urban wetlands subjected to anthropogenic pressure, located in the south of Chile.

## 1. Introduction

Wetlands are areas of land that are permanently or seasonally saturated with water and that provide numerous ecosystem services essential for supporting the lives of several organisms [1]. These ecosystems act as natural filters that purify water, remove contaminants, and act as flood controllers [2,3]. Additionally, they play a crucial role in climate regulation, helping to maintain stable temperatures and humidity levels, and they store large amounts of carbon, contributing to the mitigation of climate change effects [1,4,5].

Despite their environmental importance, wetlands have suffered dramatic declines. Since 1900, it is estimated that nearly 64% of the world’s wetlands have disappeared. This loss is due to various human activities, such as urbanization, intensive agriculture, and pollution, which have led to the degradation and disappearance of these ecosystems [6,7]. To address this crisis, the Ramsar Convention, signed in 1971, aims to conserve and wisely use wetlands through local and national actions and international cooperation [8]. Chile joined the Ramsar Convention in 1981 and currently has 16 Ramsar sites. Additionally, in 2022, Chile enacted the Urban Wetlands Law (Law 21.202), aiming to protect these valuable ecosystems located in the urban radio. In Chile, wetlands cover approximately 6% of the national territory, which represents more than 40,000 wetlands, including around 1400 urban. These wetlands not only offer ecological benefits but also provide crucial services to local communities, such as water provision, recreation, and biodiversity conservation [1]. Regardless of its numerous benefits, Chilean urban wetlands are under continuous threats and constantly subjected to anthropogenic pressure. The lack of environmental awareness and education, together with economic activities and pollution, endangers the integrity of these ecosystems. 

Zoonoses, defined as diseases transmitted from animals to humans [9], are responsible for millions of cases and deaths annually worldwide. The Pan American Health Organization (PAHO) has warned about the serious threat zoonotic diseases pose to human health. It is estimated that around 60% of the known human pathogens have a zoonotic origin [10]. In Chile, some zoonoses are monitored by government entities. For example, two of the most prevalent zoonoses in southern Chile are brucellosis and leptospirosis, caused by bacteria of the genus *Brucella* and *Leptospira*, respectively [11,12]. Many of the etiological agents of zoonotic diseases can be dispersed through water; hence, it is necessary to maintain strict control of water quality and implement an interdisciplinary approach to avoid pathogen dispersion [13]. Collaborative and interdisciplinary context seeks to sustainably balance and optimize human, animal, and ecosystem health. Recently, microbial surveillance using molecular approaches has been postulated as crucial for monitoring the presence of zoonotic pathogens in the environment [14,15]. In Chile, the Agricultural and Livestock Service (‘Servicio Agrícola Ganadero’, SAG) is the main governmental agency that monitors the health issues related to agricultural and livestock activities.

Several factors, such as habitat fragmentation, environmental pollution, and climate change, have increased the presence of emerging infectious diseases in the last century [16]. The “*Tripartite Guide to Address Zoonotic Diseases in Countries*”, published in 2021 in collaboration with the FAO, OIE, and WHO, offers guidance to address zoonotic diseases and other health emergencies, indicating that disease surveillance of zoonotic infections in the environment should be one of the main tasks to prevent the dispersion of these [17]. For example, Fresia et al. provided evidence of the prevalence of antimicrobial resistance genomic elements in coastal environments located close to Montevideo, Uruguay, including beach and wastewater samples, characterizing pathogenic bacterial communities and their virulence repertoires. The evidence shows the presence of antibiotic-resistant genes in both wastewater and beach environments [18]. Recent studies have identified several bacteria of sanitary interest in water bodies of the Los Lagos Region in Chile, highlighting the need to monitor and protect these ecosystems to prevent health risks [19,20]. For that purpose, the information recovered from DNA sequencing is an excellent tool, which provides data about taxonomic diversity and gene function [21,22]. In fact, the approach known as shotgun metagenomics sequences all genetic material in a sample, enabling analysis of microbial diversity and functions at the same time. This improves the understanding of bacterial communities and reveals functional information such as the presence of antimicrobial-resistant genes and genetic virulence mechanisms. Identifying and monitoring these genes is essential for managing and mitigating environmental biohazards, reducing the risk of disease transmission. This work investigates the presence of antimicrobial-resistant genes (ARGs) and virulence factors (VFs) in various wetland environments. By employing shotgun metagenomic and bioinformatic analysis, we detected a wide range of ARGs and VFs in water recovered from urban wetlands. Our findings suggest that impaired wetlands can serve as reservoirs for these genetic elements, potentially facilitating the spread of antibiotic resistance and accumulating microbial pathogens. This research underscores the importance of monitoring wetlands to assess and mitigate public health risks associated with antimicrobial resistance and infectious diseases.

## 2. Materials and Methods

### 2.1. Area of Study and Sample Collection

A total of 13 urban wetlands were sampled during the year 2023, located in the urban radio of the cities Osorno (wetlands namely Las Quemas and Ovejería, both sampled in 4 October 2023), Llanquihue (namely El Loto, Baquedano, Las Ranas, and Teodosio Sarao, all sampled in 22 May 2023), Puerto Varas (namely Marina and Quebrada Parque, both sampled in 6 June 2023), and Puerto Montt (namely Luis Ebel, La Paloma, Antiñir, Rupallán, and Mirasol, sampled in 13 September 2023, 3 November 2023, 29 August 2023, 6 September 2023, and 16 August 2023, respectively) of the Los Lagos region of Chile (Appendix A) were sampled. The wetlands, namely Luis Ebel and Teodosio Sarao, were sampled two times at different moments (30 October 2023 and 2 November 2023, respectively). A volume of 3 L per sample site was taken using a sterile 1 L glass jar. The samples were transported cold for subsequent processing at the Institutional Laboratory of Universidad San Sebastián, located in the city of Puerto Montt, Chile. The samples were individually filtered through MCE (mixed cellulose ester) membranes with a pore size of 0.22 µm and a diameter of 47 mm (Whatman #WHA10401712, Maidstone, UK). For this purpose, a negative pressure system was used, drawing the microorganisms through the filter. Subsequently, each filter was stored frozen at −80 °C until DNA extraction. 

### 2.2. Genomic Material Recuperation and DNA Sequencing

The stored MCE filters were used for DNA extraction employing the AccuPrep Genomic DNA Extraction Kit (Bioneer #K-3032, Daejeon, Republic of Korea), following manufacturer protocol, with few modifications. Filters were submerged in 500 μL DNA extraction buffer and stirred to release microbial cells. An enzymatic digestion was used to facilitate microbial cell wall rupture, using 20 μL of lysozyme (20 mg/mL) and 20 μL of proteinase K (20 mg/mL) [23], incubating the suspension for 1 h at 37 °C and then for 1 h at 55 °C, respectively. After enzymatic digestion, the steps provided by the manufacturer protocol for bacterial DNA extraction were followed. The obtained DNA was quantified by absorbance, and the ratios 260/280 nm were calculated to assess the purity of the DNA obtained, and its integrity was evaluated through 1% agarose gel electrophoresis. Before DNA sequencing, we test the amplification capacity of DNA using 16S bacterial universal PCR. A total of 1 μg of DNA was sent to Novogene (Sacramento, CA, USA) genomic service for shotgun metagenomic sequencing. DNA was sequenced by paired-end (2 × 150 bp) reads using the Illumina NovaSeq 6000 platform with an output of 6 GB per sample.

### 2.3. Metagenomic Data Analysis and Functional Characterization

Raw data obtained from the sequencing provider were initially inspected with FastQC [24], and then reads were filtered and trimmed using Trimmomatic [25] using the following parameters: LEADING:20, TRAILING:20, SLIDINGWINDOW:5:20, AVGQUAL:20, and MINLEN:90, followed by the application of Bowtie2 to screen out the contaminant DNA sequences from humans and viruses [26]. Filtered data were assembled using MegaHit [27], and the quality of the content obtained was inspected using Quast [28]. Taxonomic profiling was assessed using Kraken2 [29], keeping the taxonomic assignation with over 50 hits per sample, and relative abundance was assessed by dividing the hits obtained for each taxonomic assignation at the specie level by the number of total hits classified in each sample. The antimicrobial resistance and virulence genes were inspected using ABRicate [30], utilizing the ResFinder [31], NCBI AMRFinderPlus [32], and VFdb [33] databases. Mobile genetic elements were retrieved from metagenomic assembled sequences over 2000 pb using plaSquid [34] through plsdb plasmid sequence database assignation [35], and mobile sequences obtained were used to look for the presence of AMR genes carried in the mobile elements using ABRicate with both ResFinder and AMRFinderPlus databases, as described above. Data obtained were imported to R statistical language [36] for further analysis and representation using the ggplot2 package [37]. 

## 3. Results

### 3.1. Bacterial Taxonomy in Urban Wetlands

Shotgun metagenomic and bioinformatic studies indicate a heterogeneous bacterial composition and abundance between the studied urban wetlands (Figure 1 and Appendix A). A total of 6493 bacterial species were identified (Appendix A), including the genus *Pseudomonas*, *Serratia*, *Rahnella*, *Shewanella*, and *Aeromonas*, which has the most abundant species in all the wetlands analyzed. In the wetlands located in Osorno, the most abundant species were *Rahnella aceris*, *Flavobacterium ammonificans*, *Limnohabitans* sp. TEGF004, *Stenotrophomonas rhizophila*, and *Pseudomonas sivasensis*, while the abundant species in the Llanquihue wetlands were *Pseudomonas fragi*, *Pseudomonas psychrophila*, *Pseudomonas monsensis*, *Serratia proteamaculans*, and *Shewanella baltica*. Moreover, in the Puerto Varas wetland, the bacterial species with the most abundance were *Pseudomonas* sp. B21-035, *Pseudomonas protegens*, *Erwinia rhapontici*, *Rahnella aceris*, and *Serratia proteamaculans*, while in the Puerto Montt wetlands the observed species were namely *Serratia liquefaciens*, *Serratia fonticola*, *Pseudomonas fragi*, *Pseudomonas monsensis*, and *Serratia grimesii* (Appendix A). Overall, in all the wetlands, the bacteria of the genus *Pseudomonas* were observed as the most abundant species. 

### 3.2. Identification of Bacterial Microorganisms of Importance in Veterinary and Human Medicine

Bacterial species of importance in veterinary medicine were searched within the taxonomic data obtained, which are also included in the list of mandatory notifiable diseases monitored by the Agriculture Livestock Service of Chile (SAG) [38]. From the mentioned list of diseases, some etiologic agents were identified at the species level, present in the water of the urban wetlands. Some of these bacteria were identified in more than one wetland; however, the abundance of these was different (Figure 2). Among the identified species of importance to health and which are notifiable to the SAG were found to be the species *Pasteurella multocida*, *Salmonella enterica subsp. Typhimurium*, *Salmonella enterica subsp. Enteritidis*, *Chlamydia psittaci*, *Coxiella burnetii*, *Bacillus anthracis*, and *Streptococcus equi*, which are responsible for the diseases fowl cholera, salmonellosis, psittacosis, Q fever, carbuncle, and equine adenitis, respectively. 

Additionally, in the taxonomic assignation tables (Appendix A), the presence of microorganisms potentially infectious for animals and humans can be found (Figure 3). Each wetland studied contains a different relative abundance of these bacteria. Remarkably, a high diversity of *Pseudomonas aeruginosa* and *Escherichia coli* was found in all the studied wetlands. The next top bacteria observed belong mainly to *Bordetella*, *Staphylococcus*, and *Clostridium* genus, with high relevance in public health because *Bordetella* species can lead to respiratory illnesses like whooping cough, while *Staphylococcus* species can cause skin infections, while *Clostridium* species are associated with foodborne illnesses such as botulism and tetanus.

### 3.3. Detection of AMR Genes and Virulence Mechanisms on DNA Recovered from Urban Wetlands

The metagenomic analysis revealed a diverse repertory of antimicrobial-resistant genes in the environmental DNA recovered from the water of urban wetlands. The different genes identified according to the antimicrobial resistance family to which they belong were grouped, and the resistance that confers was identified (Figure 4, Appendix A). These ARGs included genes that confer resistance to antibiotics commonly used in human clinical treatments, such as β-lactams, carbapenems, and chloramphenicol, as well as genes associated with resistance to antibiotics used in veterinary medicine, such as macrolides, tetracyclines, and quinolones. The presence of these genes and the kind of genes identified were different in each urban wetland (Figure 4, Table 1, Appendix A), which may be related to the use of antibiotics based on compounds such as amoxicillin, ampicillin, ciprofloxacin, nalidixic acid, and imipenem.

Additionally, the data also show the presence of virulence mechanisms originally presented in the genomes of the bacterial strains, namely *Yersinia enterocolitica* subsp. *enterocolitica* 8081, *Pseudomonas aeruginosa* PAO1, *Escherichia* coli O157:H7 str. EDL933, *Salmonella enterica* subsp. *enterica* serovar *typhimurium* LT2, and *Shigella dysenteriae* Sd197 (Table 2, Appendix A, Appendix A). Overall, the genes associated with *Y. enterocolitica* subsp. were related to the flagellar motor proteins *fli*G and *fli*M, which are crucial proteins for the rotation of the flagellum, providing bacterial motility. Among the genes associated with *P. aeruginosa*, the elevated expression of genes related to virulence, such as the *pvd*S sigma factor of extra-cytoplasmic function and the *flg*C protein of the flagellar basal body, is related to the capacity of the bacteria to move in the media to acquire nutrients, which is key for its survival and pathogenicity. Moreover, the main virulence mechanism identified in *E*. *coli* O157 was mediated by the *yag*Z/*ecp*A genes, which encode the structural subunit EcpA of the pilus, which plays a crucial role in adherence to surfaces and interaction with host cells. Finally, *S. enterica* subsp. *enterica* serovar *Typhimurium* showed that the main virulence mechanism was encoded by the gene *csg*F, related to the assembly of curli fibers that help bacteria to adhere to surfaces and form biofilms [44]. 

### 3.4. Identified Mobile Genetic Elements in Wetlands Carrying Antimicrobial Resistance Genes

To further complete the characterization of ARGs present in the urban wetlands, metagenomic assemblies were screened to explore the presence of mobile genetic elements (MGEs). The results showed the presence of 1844 MGEs in all the wetlands analyzed, mainly identified on bacteria’s genera, namely *Rahnella*, *Serratia*, *Pseudomonas*, *Aeromonas*, and *Klebsiella*, regarding the plasmid assignation in the psldb database (Appendix A). To a lesser extent, some MGEs were annotated as belonging to the genus *Shewanella*, *Erwinia*, *Enterobacter*, *Acinetobacter*, *Escherichia*, and *Salmonella*. A total of 307 plasmid sequences were non-assigned (NA) to the plsdb database. We also observed an overlap between the original host reported for MGEs and the presence of harmful microbial species in the taxonomic assignation obtained from metagenomic data (Appendix A). To further describe MGEs, ARGs carried into the plasmid sequences were analyzed. A total of four ARGs were identified into plasmid sequences: *bla*RAHN_22 was found in wetlands Baquedano, Las Quemas, Marina, and Teodosio; *bla*EL-18 in wetland Marina; *mcr*_4.3_1 in wetland Rupallán; and *tet*(H)_3 in wetland Loto (Table 3), which confers resistance to betalactamics, macrolides, and tetracyclines, respectively.

## 4. Discussion

Accelerated population growth, together with unplanned urbanization and extensive deforestation, has significantly modified the borders between human and animal populations, considerably altering the balance of ecosystems [45]. These anthropogenic activities not only contribute to the loss of biodiversity and the destruction of natural habitats but also increase the incidence of zoonotic diseases by facilitating the impairment of the environment and stretching the contact between humans and wildlife [9]. For these reasons, it is needed to establish molecular surveillance programs that allow the detection of biohazard elements in the environment [46]. The characterization of microorganisms present in bodies of water is a research area of great relevance for public health and policy formulation. This information allows us to deeply understand local microbial diversity, identify patterns of antibiotic resistance, and establish relationships between human activity, urban infrastructure, and the well-being of nearby populations [47]. Studies like those conducted by Fresia et al. (2019) and Campanini-Salinas et al. (2024) have demonstrated the prevalence of antibiotic-resistant genes in coastal and urban environments in Uruguay and Chile, highlighting the need for comprehensive strategies to environmental surveil and avoid antimicrobial resistance transmission in both the veterinary and human health [18,20].

As mentioned by Ballesteros et al. (2023), who analyzed seven wetlands with different anthropogenic loads using metagenomic amplicon sequencing, a high abundance of genera, namely *Pseudomonas*, *Flavobacterium*, *Aeromonas*, and *Mycoplasma*, was observed, similar to the observations performed in this study [48]. These findings suggest that pathogenic microorganisms are present in high abundance in urban wetlands, which represents a potential risk to animal health, especially for those species that interact directly with these waterbodies. Farm animals, pets, and even wildlife can be exposed to pathogens through contact with contaminated water.

The present study identified the presence of microorganisms in urban wetlands that are also relevant to human health. Bacteria such as *V. cholerae*, *P. aeruginosa*, and *E. coli* O157, which can cause serious infections, also were presented in our observations. This suggests that water may act as a reservoir for these pathogenic bacteria, as has been demonstrated previously in other studies [49,50,51]. Notably, in almost all the wetlands studied, bacteria belonging to the *Pseudomonas* genus were at the top of the abundance. Moreover, the detection of antibiotic-resistant genes and the characterization of virulence mechanisms in wetlands represent a tool with great potential to monitor public health risks. As mentioned by Y. Li et al. (2024), the presence of these genes can make the effective treatment of bacterial infections difficult. Antimicrobial resistance is a growing global phenomenon, and its presence in aquatic environments such as non-conserved wetlands is particularly worrying due to its close connection with human and animal activity [52].

The detection of ARGs in wetlands, coasts, and sewage indicates the presence of a reservoir of resistance that may affect marine and terrestrial fauna and ultimately human activities [53]. This situation may be exacerbated in areas close to cities, with many water bodies and a climate characterized by rain, such as the Los Lagos region in Chile, which may contribute to the accumulation and spread of microorganisms via waterways. These reservoirs can also accumulate antibiotic compounds coming from various sources, such as agriculture, livestock, and human clinical treatments, promoting the apparition of resistant bacteria phenomena [54]. Effective management of watersheds can significantly reduce the dangers to human health that arise from the presence of microbial biohazards in the environment [53]. Reducing the use of antibiotics near water bodies during cattle raising or limiting the presence of aquaculture activities in freshwater are two examples of effective public politics on water management that can help to prevent ARG transference and pharmaceutical pollutant dispersion [55,56,57]. In addition, monitoring and regulating the discharge of wastewater from urbanization and hospitals can help reduce the load of microbial biohazards and pharmaceutical pollutants in water systems [58]. Furthermore, evidence-based decision-making on water quality, supported by monitoring data, can be critical for providing safe drinking water, optimizing water quality, and effectively managing water resources [59,60]. Preserving the water resource will protect the health of the surrounding communities and the natural environment and also help to maintain the long-term efficacy of antibiotics and sustainable public health management.

## 5. Conclusions

In this work, shotgun metagenomics was used to describe the bacterial and functional diversity present in urban wetlands, which serves as a demonstration of how metagenomics might help monitor microbial biohazards at the interface of urban emplacements that are closely connected to the natural environment. Several environmental and pathogenic bacteria were identified, and a few of them are the etiological agents of diseases listed in the mandatory notifiable diseases group, monitored by the Agriculture Livestock Service of Chile (SAG). Additionally, species with high prevalence relevant to human and animal health were detected. Functionally, ARGs and virulence mechanisms were identified, even carried in mobile genetic elements. Our study demonstrated the utility of metagenomics in identifying genetic determinants and microbial biohazards in urban wetlands subjected to anthropogenic pressure. 

## Figures and Tables

**Figure 1 pathogens-13-00788-f001:**
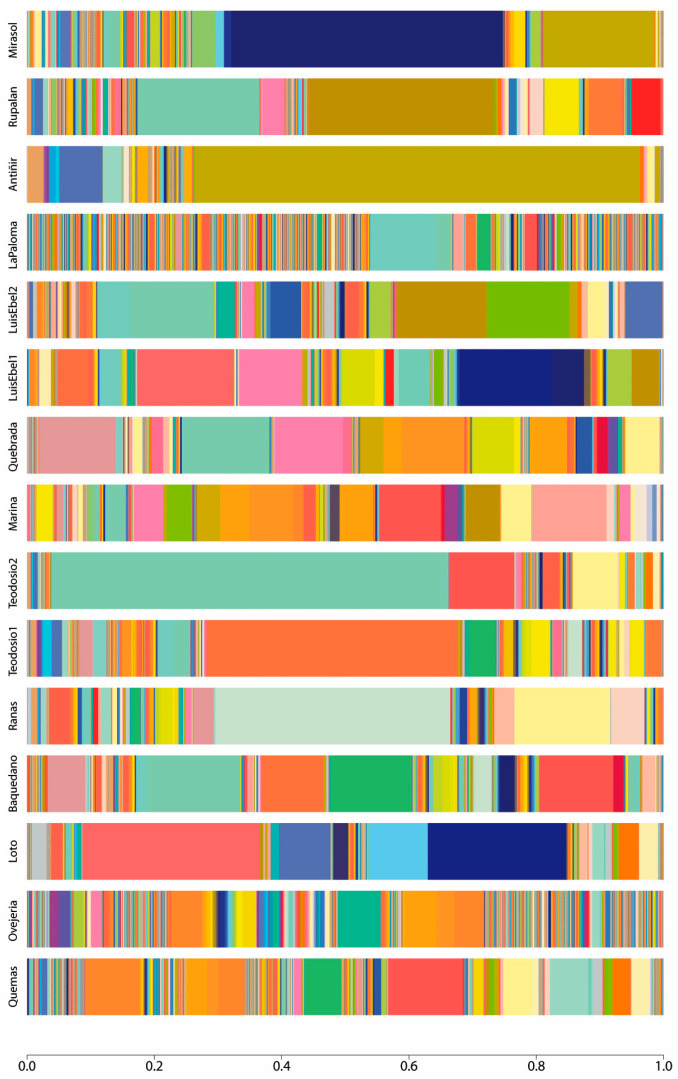
**Taxonomy abundance at the species level is represented as a stacked bar plot of each sample.** The color pattern of each bar shows the microbial community structure, while the amplitude of each color represents the percentage of abundance of the assigned taxonomy. Remarkable taxa with the most abundance was *Pseudomonas* sp. SCA2728.1_7 on the Mirasol wetland, represented by the dark blue color, *Serratia liquefaciens* on Antiñir, denoted by olive color, *Pseudomonas fragi* on Teodosio2 represented by cyan color, while on Teodosio1 the most abundant specie was *Pseudomonas psychrophile*, denoted by tomato color.

**Figure 2 pathogens-13-00788-f002:**
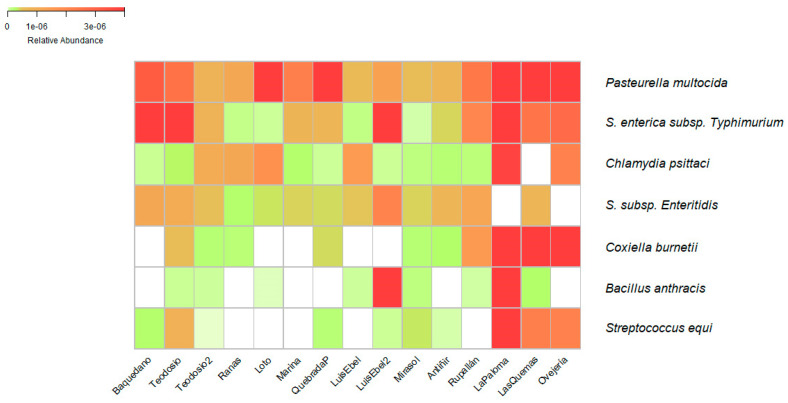
Heatmap of bacteria species identified that are included in the list of notifiable diseases. The relative abundance of bacterial species in the cities of Osorno, Llanquihue, Puerto Varas, and Puerto Montt is represented by the colors white (low abundance), yellow (middle abundance), and red (high abundance). The wetland with the highest relative abundance of microorganisms, represented by an intense red color, is the La Paloma wetland, located in the city of Puerto Montt. Conversely, the wetland with the lowest abundance is the Luis Ebel wetland, also in the city of Puerto Montt.

**Figure 3 pathogens-13-00788-f003:**
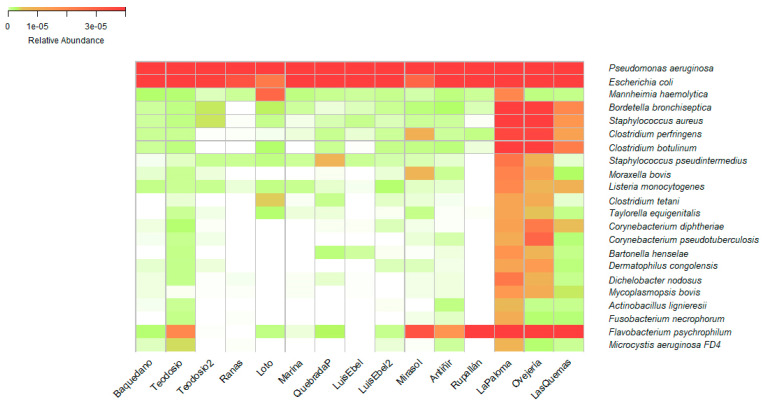
Heatmap of bacterial species relevant in veterinary and human medicine. The relative abundance of bacterial species in wetlands of the cities of Osorno, Llanquihue, Puerto Varas, and Puerto Montt is presented. The wetland with the highest relative abundance of infectious bacterial species, represented by an intense red color, is the La Paloma wetland in the city of Puerto Montt. Conversely, the wetlands with the lowest abundance are Las Ranas in the city of Llanquihue, Luis Ebel in the city of Puerto Montt, and La Marina in the city of Puerto Varas.

**Figure 4 pathogens-13-00788-f004:**
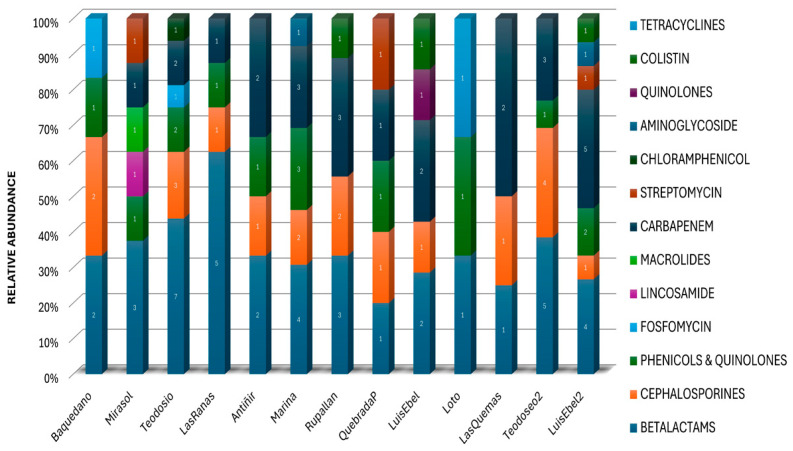
Identified antimicrobial resistances and classification by antimicrobial families or by antimicrobial compounds in urban wetlands of the Los Lagos region. A wide diversity of ARGs were identified using the NCBI database, with the highest frequency corresponding to the Teodosio Sarao wetland in the city of Llanquihue, followed by the Luis Ebel wetland in the city of Puerto Montt. The number denotes the different genes for the same resistance.

**Table 1 pathogens-13-00788-t001:** Summary of most prevalent antibiotic-resistant genes found.

Family Gene	Antibiotic Family	Drug Example	Resistance Mechanism Example	Literature References
*bla*	Betalactamics	imipenem	Antibiotic inactivation	[39]
*oqx*	Fluoroquinolones	ciprofloxacin	Antibiotic inactivation	[40]
*lnu*	Lincosamides	clindamycin	Antibiotic inactivation	[41]
*mph*	Macrolides	erythromycin	Antibiotic inactivation	[42]
*tet*	Tetracyclines	doxicycline	Efflux pump, target modification, antibiotic inactivation	[43]

**Table 2 pathogens-13-00788-t002:** Presence of virulence mechanisms associated with pathogenic bacterial species.

Gene Name	Sub-Clasification	Function	Biological Component	Bacteria
*csg*	B, D, E, F y G	Adherence	Cell wall	*S. typhimurium*
*spv*	C	Effector delivery system	Cytoplasm	*S. typhimurium*
*ste*	B	Effector delivery system	Plasma membrane	*S. typhimurium*
*alg*	I y U	Biofilm	Plasma membrane	*P. aeruginosa*
*fle*	N y Q	Motility	Cytoplasm	*P. aeruginosa*
*flg*	C, G, H y I	Motility	Flagella	*P. aeruginosa*
*mbt*	H-like	Nutritional/metabolic factor	Cell wall	*P. aeruginosa*
*pil*	G	Adherence	Plasma membrane	*P. aeruginosa*
*pvd*	H y S	Nutritional/metabolic factor	Cytoplasm	*P. aeruginosa*
*waa*	F	Immune modulation	Cytoplasm	*P. aeruginosa*
*esp*	X4 y X5	Effector delivery system	Cell wall	*E. coli* O157:H7
*yag*	V, W, X, Y y Z	Adherence	Plasma membrane	*E. coli* O157:H7
*ykg*	K	Adherence	Cell wall	*E. coli* O157:H7
*che*	Y	Motility	Flagella	*Y. enterocolitica*
*flg*	B, C, G y H	Motility	Flagella	*Y. enterocolitica*
*flh*	C y D	Motility	Flagella	*Y. enterocolitica*
*fli*	A, G, M y P	Motility	Flagella	*Y. enterocolitica*
*gsp*	C, D, F, G, H, I, J, K, L	Secretion	Cytoplasm	*S. dysenteriae*

**Table 3 pathogens-13-00788-t003:** ARGs identified in mobile genetic elements. Plasmid ID is the number code in the plsdb database, which also refers to the NCBI nucleotide database. Sequence ID corresponds to the sequence code in Appendix A. The nacteria name refers to the species where the plasmid was originally assigned into the plsdb database. Wetlands is the name of the place where the plasmid was identified, while ARGs is the name of the antimicrobial-resistant gene carried on the plasmid sequence.

Plasmid Code	Sequence ID	Bacteria	Wetland	ARGs
NZ_CP019063.1	k127_72468	*Rahnella sikkimica*	*Baquedano*	*blaRAHN-2*
NZ_CP019063.1	k127_45820	*Rahnella sikkimica*	*La Marina*	*blaRAHN-2*
NZ_CP019063.1	k127_444659	*Rahnella sikkimica*	*Las Quemas*	*blaRAHN-2*
NZ_CP019063.1	k127_59296	*Rahnella sikkimica*	*Teodosio Sarao 2*	*blaRAHN-2*
NZ_CP135245.1	k127_48286	*Escherichia coli*	*La Marina*	*blaEC-18*
NZ_CP013115.1	k127_16838	*Shewanella xiamenensis*	*Rupallán*	*mcr-4.3*
NZ_CP082144.1	k127_5281	*Acinetobacter lwoffii*	*El Loto*	*tet(H)*

## Data Availability

The raw data produced from DNA sequencing in this study were deposited in the ENA-EMBL database under the accession number PRJEB77702 (https://www.ebi.ac.uk/ena/browser/view/PRJEB77702, accessed on 18 July 2024). Metagenomic data obtained from bioinformatics analysis can be found in Appendix A.

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
