# Peer review of "Metagenomics Insight into Veterinary and Zoonotic Pathogens Identified in Urban Wetlands of Los Lagos, Chile"

_pathogens, 2024, doi:10.3390/pathogens13090788_

Round 1

Reviewer 1 Report

Comments and Suggestions for Authors

The manuscript used a shotgun metagenomics approach on filtered water samples in numerous wetlands to look for zoonotic pathogens and AMR genes.  I have a few general comments.

1) The study design could use more detail - for instance it says they sampled at two different times, however where these across seasons or years?  Were the samples treated independently - like did the ask if they fond different things at different sample times or were they just merged for all analysis?

2) There are no methods explaining how they calculated relative abundance showin in Figure 2 -  Is it by read number?  Is it normalised to the amount of DNA in a sample?  Is it comparable between samples?  Without knowing how it was calculated I cannot interpret the findings.

3) They used a pipeline to find virulence genes in some of the bacteria - and state which bacteria those virulence genes were found in.  However they used a similar ppipline for the ARG genes - however they never state which bacteria contained which ARG genes. For instance in Figure 4 - are you able to identify which bacteria contained which ARG genes?  That would be more interesting and help you think about instrinsic resistance vs. acquired resistance.  As in some environmental bacteria have innate resistance genes, and so knowing if these genes are in bacteria that don't have innate resistance would be useful.  Additionally can the pipeline tell them if the gene was found on a plasma or moveable element vs. in the genome?  That would also make the findings easier to interpret.  Without discussing those items I don't think we can hypothesize that these genes are due to "antibiotic usage" so some of that language would need to be toned down.  

Author Response

Response to reviewer 1 (our answers are in red).

1) The study design could use more detail - for instance it says they sampled at two different times, however where these across seasons or years?  Were the samples treated independently - like did the ask if they fond different things at different sample times or were they just merged for all analysis?

Re: We thank the reviewer for their thoughtful comments. We took the samples across the 2023 year. We agree that there may be differences in the microbial composition according to the season of the year in which the samples were taken, however, in the Los Lagos region the seasons have a strong tendency to be rainy throughout the year. The average precipitation is close to 1100 mm of rain per year, between 56 to 176 mm per month, raining around 11 to 20 days per month. Therefore, there were no major climatic differences between the months in which the samples were taken. The samples were taken using the same procedure and were stored until DNA extraction to be processed at the same time. We improve this description and include the date of the sample in the material and methods section 2.1. Only 2 wetlands were sampled a second time on a different date, which is denoted as 2 in the sample name.

2) There are no methods explaining how they calculated relative abundance show in in Figure 2 -  Is it by read number?  Is it normalised to the amount of DNA in a sample?  Is it comparable between samples?  Without knowing how it was calculated I cannot interpret the findings.

Re: We thank the reviewer for pointing out this observation. We estimate the relative abundance using the Kraken 2 information, dividing the hits obtained for each taxonomic assignation at the species level by the number of total hits classified in each sample at the species level. This explanation was added to the Material and Methods section to improve the description of data analysis.

3) They used a pipeline to find virulence genes in some of the bacteria - and state which bacteria those virulence genes were found in.  However they used a similar ppipline for the ARG genes - however they never state which bacteria contained which ARG genes. For instance in Figure 4 - are you able to identify which bacteria contained which ARG genes?  That would be more interesting and help you think about instrinsic resistance vs. acquired resistance.  As in some environmental bacteria have innate resistance genes, and so knowing if these genes are in bacteria that don't have innate resistance would be useful.  Additionally can the pipeline tell them if the gene was found on a plasma or moveable element vs. in the genome?  That would also make the findings easier to interpret.  Without discussing those items I don't think we can hypothesize that these genes are due to "antibiotic usage" so some of that language would need to be toned down.  

Re: We thank the revisor for pointing attention to this. We agree with the reviewer in that it would be interesting to know the specific microorganisms in which these elements are present, but this is difficult to do bioinformatically with the tools that we have. We are grateful that the reviewer suggested a tool or pipeline capable of doing something like what he proposes for future studies. In this work, we use ABRicate tool for virulence and AMRg characterization which relies on local BLAST, so, the identification depends on the database used. It is possible to identify the presence of these genes, but using these tools and databases we cannot differentiate in which specific microorganisms the DNA sequence belongs, because the search is based on sequenced DNA fragments, not in complete genomes. For that reason, we indicate that these virulence genes are originally presented in the bacterial genomes of some species, taxas which also was detected in the metagenome taxonomic characterization, but we do not state which bacteria those virulence genes were found in, as the reviewer point (lines 228-229). However, with this metagenomic data, we cannot demonstrate that de AMRg and virulence genes that we detect are inside the genomes of the same bacterial species in which the database is annotated. For that purpose, we need the assembled genomes of that’s genomes, maybe using Metagenome-assembled genomes (MAGs) we could get closer to that information, but this would be outside the focus of this work, and it is a very good idea to continue our study in a new stage for a next article. However, MAGs have certain limitations, and with second-generation sequencing with 150 bp fragments, it is difficult to assemble complete large genomes. To get closer to an answer to what was raised by the reviewer, we use plaSquid bioinformatic tool, to retrieve and capture the moveable elements sequences such as plasmids, and explore the AMRg that may carry on them. This was added in section 3.4 of the results. Unfortunately, we only can indicate in which microorganisms were described that plasmid regards the database annotation, but cannot indicate inside which microbial species the plasmids were presented in our metagenomic DNA sequences.

Reviewer 2 Report

Comments and Suggestions for Authors

The manuscript entitled „ Metagenomics Insight of Veterinary and Zoonotic Pathogens  Identified On Urban Wetlands of Los Lagos, Chile” describes investigations of bacterial pathogens as well as antibiotic resistance determinants. Topic of this manuscript is an important issue, however, some minor modifications aer needed in the text.

Comments

1. Abstract is very general. Some specific data about major detected pathogens as well as antibiotic resistance determinants (genes) should be given.

2. On figure 1 each color should be indicated. Or at least the major groups that occur frequnty should be named!

3. Bacterial names must be written uniformly in all over the manuscript. „Salmonella enterica subsp Enteritidis” is written in the text, however, on figure 2 „S. subsp Enteritidis” is written.

4. Figure 4. „Antimicrobial resistances identified classify by…” proper form: Identified antimicrobial resistances and classified by …

5. Next to (or under) Figure 4 a table about the most frequent resistance genes would be useful.

6. Table 1:

„Función” Do you mean „Function”?

S. entérica tiphymurium „ Please, use proper name, or short form: S. Typhimurium

Comments on the Quality of English Language

Quality of English is good.

Author Response

 Response to reviewer 2 (our answers are in red)

  1. Abstract is very general. Some specific data about major detected pathogens as well as antibiotic resistance determinants (genes) should be given.

Re: We thank the reviewer for raising this observation. We rewrote the abstract according to your suggestion, including remarkable results and conclusions.

  1. On figure 1 each color should be indicated. Or at least the major groups that occur frequnty should be named!

Re: We thank the revisor for pointing attention to this. This kind of stacked barplot is used to show the taxonomical patterns between conditions, allowing us to represent a high density of information to contrast differences visually. We improve the description of the figure to explain the meaning of the colors and the sample description. Also, we include a description of the dominant bacteria, to denote the major groups in each sample.

  1. Bacterial names must be written uniformly in all over the manuscript. „Salmonella enterica subsp Enteritidis” is written in the text, however, on figure 2 „S. subsp Enteritidis” is written.

Re: Thanks for pointing out this misspelled. We already performed this correction in all the thext. 

  1. Figure 4. „Antimicrobial resistances identified classify by…” proper form: Identified antimicrobial resistances and classified by …

 Re: We thanks to the reviewer for this correction. The change has been made.

  1. Next to (or under) Figure 4 a table about the most frequent resistance genes would be useful.

Re: We thank to the reviewer for this suggestion. We include a table with this info as summary after firgure 4. 

  1. Table 1:

„Función” Do you mean „Function”?

S. entérica tiphymurium „ Please, use proper name, or short form: S. Typhimurium

Re: We thank the revisor for point attention to this. We rework the table to correct the misspelled errors.

Round 2

Reviewer 1 Report

Comments and Suggestions for Authors

The authors have incorporated my previous feedback and I have no further feedback to give.